# Synthesis and Study on Ni-Co Phosphite/Activated Carbon Fabric Composited Materials with Controllable Nano-Structure for Hybrid Super-Capacitor Applications

**DOI:** 10.3390/nano11071649

**Published:** 2021-06-23

**Authors:** Dalai Jin, Jiamin Zhou, Tianpeng Yang, Saisai Li, Lina Wang, Yurong Cai, Longcheng Wang

**Affiliations:** 1Key Laboratory of Advanced Textile Materials and Manufacturing Technology, Ministry of Education, Zhejiang Sci-Tech University, Xiasha Town, Hangzhou 310018, China; jdl_zist@126.com (D.J.); lnwang@zstu.edu.cn (L.W.); 2School of Materials Science and Engineering, Zhejiang Sci-Tech University, Xiasha Town, Hangzhou 310018, China; zhoujiamin321088@163.com (J.Z.); ytp147258@126.com (T.Y.); lss23901792@163.com (S.L.); caiyr@zstu.edu.cn (Y.C.)

**Keywords:** nickel-cobalt, synthetic conditions, nanostructure, flexible electrode, hybrid super-capacitor

## Abstract

The advantage of low resistivity and inactive binders makes binder-free electrode an excellent candidate for high-performance energy devices. A simple hydrothermal method was used to fabricate M_11_(HPO_3_)_8_(OH)_6_ (M: Ni and Co) (MHP) arrays combined with activated carbon fabric (ACF) without binder. The structures of MHP can be easily tuned from bouquets to nano-sheets by the concentration of NaH_2_PO_2_. The MHP/ACF composite materials with different structures showed the typical battery-type characteristic of anodic electrodes. In a three-electrode cell configuration, the MHP nano-sheet arrays/ACF composite has a higher capacity, of 1254 F/g, at a scan rate of 10 mA/cm^2^ and shows better cycling stability: 84.3% remaining specific capacity after 1000 cycles of charge-discharge measurement. The composite is highly flexible, with almost the same electrochemical performance under stretching mode. The MHP/ACF composite@ACF hybrid supercapacitor can deliver the highest energy density, of 34.1 Wh·kg^−^^1^, and a power density of 722 W·kg^−^^1^ at 1 A·g^−^^1^. As indicated by the results, MHP/ACF composite materials are excellent binder-free electrodes, candidates for flexible high-performance hybrid super-capacitor devices.

## 1. Introduction

In recent years, high-efficiency energy-storage systems and devices (photovoltaic devices, lithium ionic batteries, and other renewable energy-storage systems, etc.) have attracted lots of attention. In particular, electrochemical super-capacitors (ESCS) have shown great potential for energy-storage system. They have the advantages of a high charge and discharge rate, excellent stability, high power density, and so on [1,2,3,4,5]. Transition-metal-based compounds have been taken as promising materials for energy-storage, due to the advantages of low cost and high electrochemical activity [6,7,8].

Great efforts have been dedicated to proving that bi-metallic compounds have better energy storage performance [9,10,11]. It is reported that NiCo_2_O_4_ possesses an electronic conductivity that is higher than that of a single component metal oxide by at least two orders of magnitude [12]. Transition metal phosphites are a series of materials first reported in 1993, considered to have potential application values in many fields, like ion-exchange and catalysis, due to their open-framework structures [13]. Many groups reported the preparation of transition metal phosphites with certain morphologies, such as NiHPO_3_·H_2_O nano-needle bundles, Co_11_(HPO_3_)_8_(OH)_6_ with flower-like structures, or Ni_11_(HPO_3_)_8_(OH)_6_ nano-rods [14,15,16]. However, a simple strategy for the synthesis of bi-metallic phosphites is still a challenge. Furthermore, the preparation of transition metal phosphite arrays on a substrate is a new challenge.

Ni/Co phosphites nano-particles on conductive substrates such as carbon fabric, graphene, and carbon nanotube were reported to have a superior electrochemical performance [17,18]. Many reports considered that an ordered 1D nanostructure is particularly favored as an ideal structure for electrode materials because of its fast ion diffusion capability and large surface area [19]. In this work, Ni/Co phosphite (MHP, M = Co + Ni) arrays are designed to be deposited on activated carbon fabrics (ACF) to form a composite material. Compared with traditional metal-foil based material, ACF has the advantages of low cost, excellent electrical conductivity, and large specific surface area, which make it one of the most appreciated substrates for self-supporting electrochemical active materials. Furthermore, ACF’s flexibility is suitable for flexible electronic devices [20,21]. As reported, the performance of a micro/nano-structured super-capacitor highly depends on the size, shape, and distribution of particles [22,23]. So far, it is still a great challenge for the morphology tailoring of MHP arrays on ACF substrates.

Herein, MHP arrays were synthesized on ACF by a simple hydrothermal method, which, to our knowledge, has not been reported by other research groups. It is found that the morphology of MHP can be easily tuned from bouquet-like to nano-sheet-like by controlling the concentration of the reactants. Compared with the conventional electrode, the binder-free MHP arrays/ACF composite electrode exhibits an excellent electrochemical performance. The MHP nano-sheet arrays/ACF electrode has a specific capacity of 1254 F/g at 10 mA/cm^2^ and a 84.3% remaining specific capacity after 1000 cycles. The MHP/ACF composite@ACF hybrid supercapacitor exhibits an energy density of 34.1 Wh·kg^−^^1^ and a power density of 722 W·kg^−^^1^ at 1 A·g^−^^1^.

## 2. Materials and Methods

### 2.1. Synthesis of Carbon Fabric (ACF)

100% cotton fabric was purchased from the market (Hangzhou, Zhejiang, China). It was cleaned and soaked in 1 M NaF solution at 75 °C for 2 h. Then, the cotton fabric was dried and carbonized at high temperature in a tube furnace in argon atmosphere. The heating rate was set as 3 °C·min^−1^. The temperature was increased from room temperature to 350 °C for 30 min and then increased to 1000 °C for 1 h. The carbonized cotton fabric was cleaned and dried to achieve ACF [24].

### 2.2. Synthesis of MHP/ACF Composite Electrode

MHP/ACF composites were prepared by a one-step hydrothermal method. One millimole Ni(NO_3_)_2_·6H_2_O, 1 mmol Co(NO_3_)_2_·6H_2_O, and 2 or 4 mmol NaH_2_PO_2_·H_2_O were dissolved in 10 mL deionized water/DMAC (dimethylacetamide) (1:1). After 20 min of ultra-sonication (KH5200E ultrasonic cleaner, Hechuang Ultrasonic Instrument Co., Ltd., Kunshan, Jiangsu, China), the solution was put into a stainless steel autoclave (25 mL). A 1 cm × 1 cm ACF fabric was placed in the solution. The autoclave was heated to 160 °C for 16 h. After the autoclaves cooled down to room temperature, the ACF with active substance was taken out, cleaned, and then dried at 60 °C. MHP/ACF composites with different morphologies were achieved: the bouquet arrays (2 mmol NaH_2_PO_2_·H_2_O) and the nano-sheet arrays (4 mmol NaH_2_PO_2_·H_2_O). The weight load density of the electrodes was 0.017 g/cm^2^ (bouquet arrays) and 0.018 g/cm^2^ (nano-sheet arrays), respectively.

### 2.3. Characterizations

The crystal structure of as-prepared samples was performed on a Burke D8 X-ray diffractometer with CuK_α_ irradiation (λ: 0.154 nm). The elemental analysis was studied by EDS (electron dispersive spectroscopy). SEM (scanning electron microscopy) (ULTRA 55, Carl Zeiss SMT Pte Ltd., Germany) was used to characterize the morphology, crystal size, and intrinsic structure of all samples. A tensile test was conducted by an INSTRON 3363 electronic universal material testing machine (Xusai Instrument Co., Ltd., Shanghai, China), with a gauge length of 40 mm and a loading speed at 2 mm/min, respectively, along a 90° direction. The samples for the tensile test were cut with 15 mm in length and 10 mm in width.

### 2.4. Electrochemical Measurements

A CHI660A electrochemical workstation was used for the galvanostatic charge-discharge (GCD), cyclic voltammetry (CV), and electrochemical impedance spectroscopy (EIS) measurements. A voltage of 5 mV AC (alternating current) was applied for the EIS measurements (frequency range: 0.01 Hz to 100 kHz). The electrochemical measurements of MHP/ACF composites electrodes were performed in a three-electrode cell (WE: MHP/ACF composites electrode; CE: a platinum foil; RE: Ag/AgCl/saturated KCl). We used 3 M KOH as electrolyte solution. For the hybrid device, the MHP/ACF composite was considered as the positive and the ACF as negative electrode, respectively. We used 1 M KOH as electrolyte.

## 3. Results

As shown in Figure 1, samples with different amounts of NaH_2_PO_2_·H_2_O have different morphologies. The uniform bouquet-like morphology in Figure 1a was achieved by the addition of 2 mmol of NaH_2_PO_2_·H_2_O. The bouquet-like arrays grow radially on the ACF surface with 1–2 μm in length and about 500 nm in diameter. Each bundle of bouquet consists of several rods. Figure 1b shows the homogeneous nano-sheet morphology achieved by the addition of 4 mmol of NaH_2_PO_2_·H_2_O. The sheets are about 1–2 μm in width and 50 nm in thickness. The cross-section images in Figure 1c,d show that both arrays grew vertically on the surface of ACF.

As shown, the morphology of MHP highly depends on the concentration of NaH_2_PO_2_·H_2_O. DMAC in solution provides a basic environment to favor the dismutation of H_2_PO_2_^−^ ions to HPO_3_^2−^ ions, which may affect the crystal growth by coordinating with Ni^2+^ ions and Co^2+^ ions [25]. When a small amount of NaH_2_PO_2_·H_2_O is used, a low concentration of the HPO_3_^2−^ ion results in a slow growth of the crystal nucleus and favors the formation of a bouquet-like morphology. As the amount of NaH_2_PO_2_·H_2_O is increased, the larger amount of HPO_3_^2−^ ions might result in more nuclei and a higher crystal growth rate, which, in turn, causes the formation of nano-sheets. The bouquet or nano-sheet structure may create a sufficient contact area between the MHP/ACF electrode and electrolyte, which may lead to an excellent electrochemical performance [26].

As shown in Figure 2, XRD patterns reveal the crystal structures of as-prepared samples. It can be seen that both bouquet-like and nano-sheet like-arrays have similar diffraction patterns, which is in agreement with the monoclinic structure of Co_11_(HPO_3_)_8_(OH)_6_ (JCPDS NO. 81-1064) and Ni_11_(HPO_3_)_8_(OH)_6_ (JCPDS NO. 81-1065). The slight shift of diffraction peaks might be caused by the co-existence of Co and Ni, which indicates the formation of a monoclinic Co_x_Ni_11−x_(HPO_3_)_8_(OH)_6_ phase.

The elemental composition and distribution of MHP/ACF composites were analyzed by EDS. As shown in Figure 3a,b, the elements Ni, Co, O, and P uniformly distribute in the two samples with different morphologies. Combined with the XRD (x-ray diffraction) patterns, it can be inferred that the samples are MHP, with Ni and Co replacing each other at the same lattice position. The element carbon comes from the ACF substrates. The (Co + Ni)/P molar ratios of bouquet arrays and nano-sheet arrays are 1.5 and 1.4, which are close to the atomic ratio of M_11_(HPO_3_)_8_(OH)_6_ (11:8 = 1.375).

The electrochemical behaviors of MHP/ACF composites were studied by CV, GCD, and EIS. CV curves of MHP/ACF composites with different morphologies are given in Figure 4a,b. The oxidation peak and the reduction peak gradually shift to right and left with the increasing scan rate, respectively. The shift of redox peaks is mainly due to the fact that the internal resistance of the active material increases as the scan rates increase. In Figure 4c, the integrated area of the CV curve of MHP nano-sheet arrays/ACF sample is larger than that of MHP bouquet arrays/ACF sample. This implies that the composite electrode composed of nano-sheet arrays has a better electrochemical performance. The redox peaks correspond to the Co^2+^/Co^3+^ and Ni^2+^/Ni^3+^ transitions in the MHP nano-structure, which demonstrates the cell-type behavior of cobalt/nickel phosphite. The redox reactions might be as follows [13]:$${\mathrm{Co}}_{11}^{\mathrm{II}}{\left({\mathrm{HPO}}_{3}\right)}_{8}{\left(\mathrm{OH}\right)}_{6}+{\mathrm{OH}}^{-}↔{\mathrm{Co}}_{11}^{\mathrm{III}}{\left({\mathrm{HPO}}_{3}\right)}_{8}{\left(\mathrm{OH}\right)}_{7}+e^{-}$$
$${\mathrm{Ni}}_{11}^{\mathrm{II}}{\left({\mathrm{HPO}}_{3}\right)}_{8}{\left(\mathrm{OH}\right)}_{6}+{\mathrm{OH}}^{-}↔{\mathrm{Ni}}_{11}^{\mathrm{III}}{\left({\mathrm{HPO}}_{3}\right)}_{8}{\left(\mathrm{OH}\right)}_{7}+e^{-}$$

GCD curves are shown in Figure 5. The plateau of the GCD curve corresponds to the redox peak of the CV curve. The specific capacity was calculated according to following equations [27]:(1)$$C_{s}=\frac{2i_{s}\times \int^{\;}Vdt}{V{/}_{V_{i}}^{V_{f}}}$$
(2)$$C_{m}=\frac{C_{s}\times S}{m}$$

In Equation (1), *C_S_* (F·cm^−2^) represents the real special capacitance, *i_S_ = I/S* (A·cm^−2^) is the current density, *I* (A) is the current, *S* (cm^2^) is the area of working electrode, *∫Vdt* (V·s) is the integral current area, *V* (V) is the potential with initial and final values of *V**_i_* (V) and *V**_f_* (V), respectively.

In Equation (2), *C_m_* (F·g^−1^) represents the galvanostatic specific capacitance, and m (g) is the mass loading of the working electrode.

The calculated specific capacity is given in Figure 5c. As current density increases, the ions in the electrolytes do not have enough time to diffuse into the electrode material and the redox reaction does not occur sufficiently. In addition, the overpotential caused by physical resistance and electrochemical polarization becomes larger, which causes a larger actual difference between the charging state of the electrode and the terminal voltage and leads to a reduction in capacity. As a result, increasing current density leads to a rapid decrease of specific capacity with capacity retention of 31.7% (nano-sheet arrays) and 28.5% (bouquet arrays), respectively. The GCD results demonstrate that composite electrodes with nanosheet arrays have a better electrochemical performance. However, this electrode material has a poor electrochemical performance in a high current density operation environment.

The kinetic analysis is performed to study the relationship between the current at a particular potential and scan rate. The relation of *ν* (scan rate) and *i_p_* (peak current) is studied by Equation (3) [28,29] to analyze the mechanism of charge storage and the reaction dynamics of the composite electrode:(3)$$i_{p}={a\nu }^{b}$$

According to Equation (3), *b* is the slope of the plot log(*i_p_*) vs. log(*ν*). The property of charge storage process is surface-limited (capacitive type) or diffusion-controlled (battery type) as *b* is equal to 1 or 0.5, respectively. The MHP/ACF electrode with bouquet arrays has *b* values of 0.528 and 0.519 for anodic and cathodic scans, as shown in Figure 6a. The MHP/ACF electrode with nano-sheet arrays has *b* values of 0.381 and 0.388, as shown in Figure 6d. Apparently, both composite electrodes show battery-type behavior [30].

As battery-type materials with nanostructure generally have a large surface volume ratio, certain non-diffusion-related charge storage processes should be observed, especially at high scanning rates. Therefore, *i_V_* (current response at given potential) may be expressed by Equation (4) [31,32]:(4)$$i_{V}{=k}_{1}{\nu +k}_{2}\nu^{1/2}$$
where *k*_1_*ν* refers to surface-confined current and *k*_2_*ν*^1/2^ refers to diffusion-controlled current. Herein, *k*_1_ and *k*_2_ can be achieved from the slope of the *i*-*v* curve and *i*-*v*^1/2^curve, respectively. Then, the plot of *k*_1_*ν* as a function of *v* (potential) can be drawn by fitting the CV curve. The blue region in Figure 6b shows the fitting result for the MHP/ACF composite with bouquet arrays at a scan rate of 5 mV/s. The ratio of the blue region area to the enclosed area of the CV curve suggests that the surface constraint contribution to the total capacity is 42.51%. The surface constraint contribution increases with the increasing scan rate, as shown in Figure 6c, which indicates that the charge-storage process is dominated by surface/near-surface related behavior at a high scanning rate [33]. While as shown in Figure 6e,f, the MHP/ACF composite with nano-sheet arrays exhibits a relatively lower surface constraint contribution. This may be the reason for its higher performance of charge storage.

EIS characterization was used to study the impedance performance of the MHP/ACF composite, and the results are shown in Figure 7a. As shown, the linear plots in the low frequency region are related to the diffusion resistance (Warburg impedance), which represents the diffusion of electrolyte in the porous structure of the electrode material and the diffusion of protons in the substrate material. Nano-sheet array electrode materials have a high curve slope and good kinetics, which are more favorable for ion transport [34]. The intersection of the curves at the real part Z’(R_s_) in the high frequency region shows the combined resistance of contact resistance, electrolyte ion resistance, and electrode material intrinsic resistance between the active material and the collector interface. The small R_s_ of nano-sheet arrays and bouquet arrays electrode materials indicates the low intrinsic resistance of the electrode material and the low contact resistance. Similarly, the charge transfer resistance (R_ct_) of the nano-sheet arrays (2.8 Ω) and nano-bouquet arrays (3.2 Ω) is also small, which indicates that the active material grows directly on the surface of the substrate material with a low interfacial charge transfer resistance [35]. As indicated by the results in Figure 7b, the MHP/ACF composite with nano-sheet arrays exhibits a better cycling stability: 84.3% remaining specific capacity after 1000 cycles (82.4% for the bouquet-like arrays). After long cycling, repeated volume deformation generates internal osmotic stress leading to active material detachment; the coalescence of nanoparticles and the gradual collapse of the array structure may be the reasons for the reduction of active sites and the deterioration of electrochemical properties [36].

The obtained MHP/ACF composite is highly flexible, and can be stretched, twisted, and rolled up repeatedly as shown in the inset of Figure 8a and Appendix A. A tensile test was further carried out to test the stress-strain deformation behavior of the composite. The shape of the stress–strain curve of the MHP/ACF composite in Figure 8a is similar to our previous work [24], with a maximum load of 3.1 MPa and a considerable strain of more than 5%. The electrochemical properties of the MHP/ACF composites with nano-sheet arrays with 20% stretching and without stretching are shown in Figure 8b. Almost the same electrochemical performance was achieved in tensile mode, which indicates the potential applications of the MHP/ACF composite as a flexible electrode.

A two-electrode electrochemical cell was fabricated to fully explore the electrochemical performance of the MHP nano-sheet arrays/ACF composite. The positive electrode was an MHP nano-sheet arrays/ACF composite, the negative electrode was ACF, and the electrolyte was 1 M KOH. The CV curves of the hybrid device at different scan rates, from 10 to 50 mV·s^−1^, are shown in Figure 9a. All the curves display a quasi-rectangular profile as a result of a collective influence of EDLC (electric double layer capacitor) (ACF) and faradaic (MHP) activities, which is a distinctive performance of the hybrid supercapacitor. GCD curves at different specific currents, from 1 to 10 A·cm^−1^, are shown in Figure 9b. The specific energy density and the power density were calculated from the GCD curves according to the following equations [37]:(5)$$E_{d}=\frac{i\;}{3.6\times \left(m_{+}+m_{-}\right)}\int^{\;}Vdt$$
(6)$$P_{d}=3600\;\times \;\frac{E_{d}\;}{\Delta t}$$

In Equations (5) and (6), *E_d_* is the specific energy density (Wh·kg^−^^1^), *P_d_* is the specific power density (W·kg^−^^1^), *i* is the applied current (mA), *m* is the total mass of the active material (mg), $\int^{\;}Vdt$ is the area under the discharge curve of the device, and Δ*t* is the discharge time (s). Considering that ACF also contributes to electronic storage, it is regarded as an active material, so that m_+_ is the mass of the MHP/ACF composite electrode, which is 0.0753 g; and m_-_ is the mass of the ACF used as negative electrode, which is 0.0500 g.

The Ragone plot of the MHP/ACF@ACF hybrid device and some results from other works are listed in Figure 9c. The MHP/ACF@ACF hybrid device recorded the highest energy density, of 34.1 Wh·kg^−1^, and a power density of 722 W·kg^−1^ at 1 A·g^−1^, which is similar or superior to those reported results [13,38,39,40,41,42,43].

## 4. Conclusions

In summary, a simple hydrothermal method was adopted to synthesize MHP/ACF composite materials with tunable array-structure for the applications of flexible high-performance super-capacitors. The structure of the MHP/ACF composite material could be simply tuned from bouquet-like to nano-sheet-like by the concentration of NaH_2_PO_2_. MHP/ACF composite materials, especially the nano-sheet-like MHP/ACF electrode, show an excellent electrochemical performance. The nano-sheet like MHP/ACF electrode has a higher specific capacity (1254 F·g^−1^, 10 mA·cm^−2^) and better long-term stability (84.3% of capacity maintained after 1000 cycles). The hybrid electrochemical capacitor device fabricated with MHP nano-sheet arrays/ACF as the positive electrode and ACF as the negative electrode attained an energy density of 34.1 Wh·kg^−^^1^ and a power density a of 722 W·kg^−^^1^ at 1 A·g^−^^1^, respectively. The unique nanostructure—nano-sheets directly grown on the activated carbon fabric (ACF), which has no additives and binders with higher specific surface areas and electrical conductivity—might be the major factor for the excellent electrochemical performance. As indicated by the results, the MHP/ACF composite material with a tunable nanostructure might be a promising candidate material for a high-performance flexible electrode for energy-storage technology.

## Figures and Tables

**Figure 1 nanomaterials-11-01649-f001:**
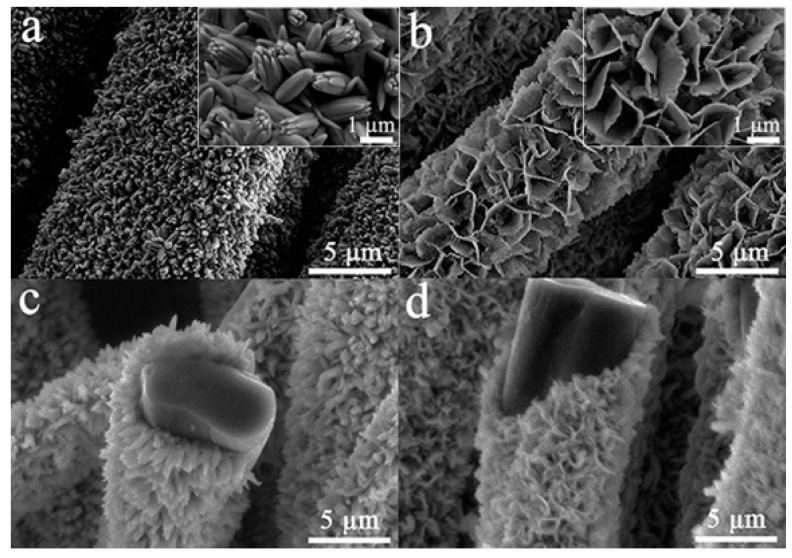
Surface morphology of as-prepared samples: (**a**) MHP bouquet arrays/ACF (the inset figure is magnified image of given sample); (**b**) MHP nano-sheet arrays/ACF (the inset figure is magnified image of given sample); (**c**) the image of MHP bouquet arrays/ACF’s cross section; (**d**) the image of MHP nano-sheet arrays/ACF’s cross section.

**Figure 2 nanomaterials-11-01649-f002:**
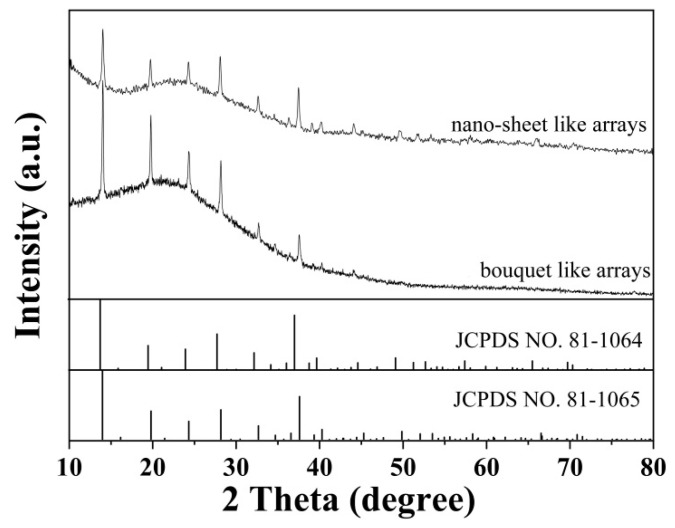
XRD patterns of as-prepared samples of MHP arrays on ACF.

**Figure 3 nanomaterials-11-01649-f003:**
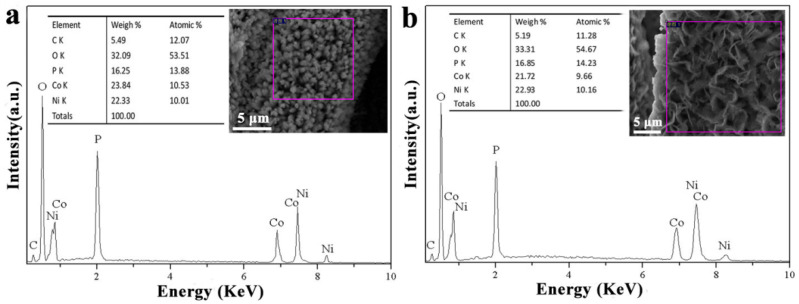
EDS curves of (**a**) MHP bouquet arrays/ACF (inset figure is the selected area for EDS characterization); (**b**) MHP nano-sheet arrays/ACF (inset figure is the selected area for EDS characterization).

**Figure 4 nanomaterials-11-01649-f004:**
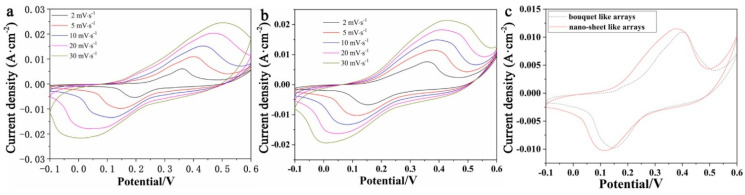
CV curves of (**a**) MHP bouquet arrays/ACF electrode; (**b**) MHP nano-sheet arrays/ACF electrode; (**c**) both electrodes at scan rate of 5 mV/s.

**Figure 5 nanomaterials-11-01649-f005:**
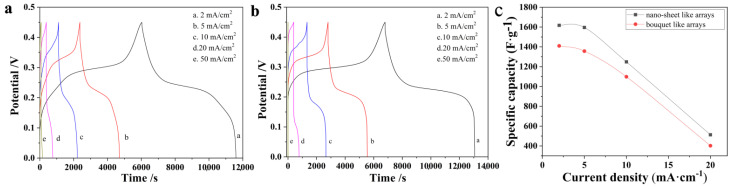
GCD curves of (**a**) MHP bouquet arrays/ACF composite electrode; (**b**) MHP nano-sheet arrays/ACF composite electrode; (**c**) Specific capacity of MHP/ACF composite electrode.

**Figure 6 nanomaterials-11-01649-f006:**
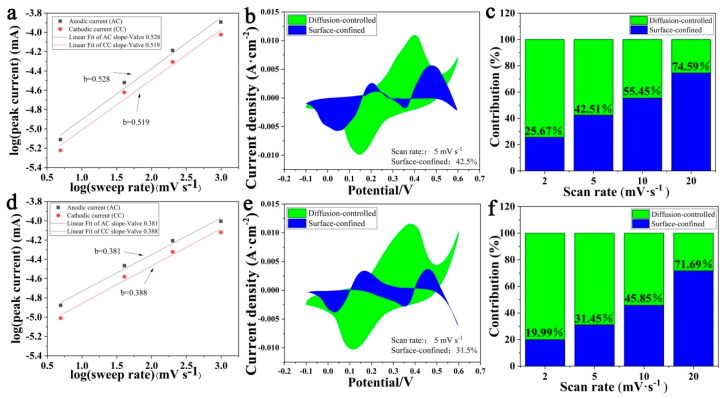
MHP bouquet arrays/ACF: (**a**) log(i_p_) vs. log(ν); (**b**) different contributions to charge storage: surface-confined (blue) and diffusion-controlled (green); (**c**) The ratios of diffusion-controlled contribution and surface-confined contribution at different scan rates. MHP nano-sheet arrays/ACF: (**d**) log(i_p_) vs. log(ν); (**e**) different contributions to charge storage: surface-confined (blue) and diffusion-controlled (green); (**f**) The ratios of diffusion-controlled contribution and surface-confined contribution at different scan rates.

**Figure 7 nanomaterials-11-01649-f007:**
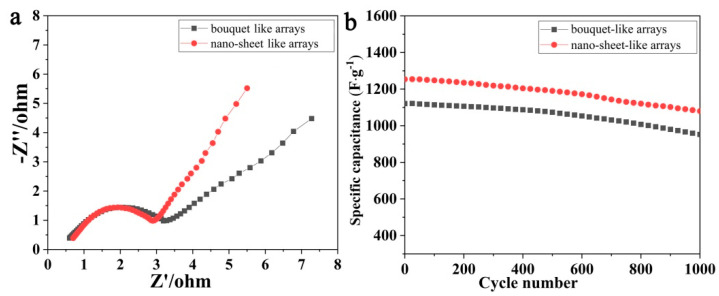
(**a**) Nyquist plots of MHP/ACF electrodes; (**b**) Cycling properties of MHP/ACF electrodes (current density: 10 mA·cm^−^^2^).

**Figure 8 nanomaterials-11-01649-f008:**
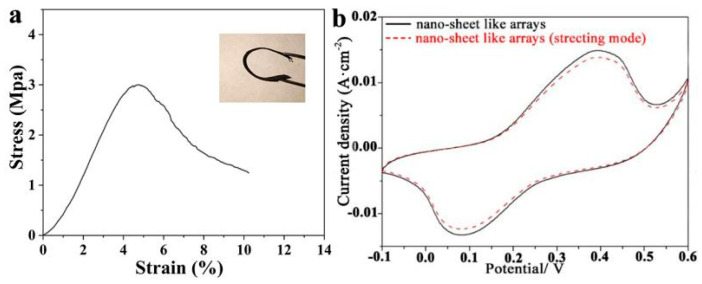
(**a**) Stress-strain curves of the MHP nano-sheet arrays/ACF composite (the inset figure is the digital image of bended sample); (**b**) CV curves of MHP nano-sheet arrays/ACF composite electrode before and after stretching.

**Figure 9 nanomaterials-11-01649-f009:**
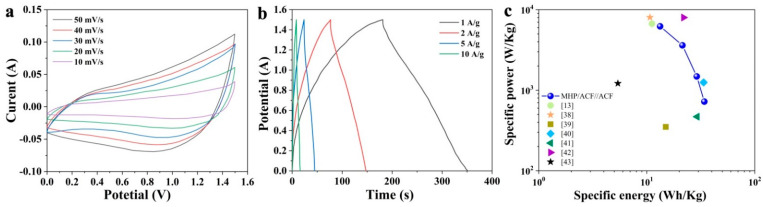
(**a**) CV curves of the MHP nano-sheet arrays/ACF composite//AC hybrid device measured within 0–1.4 V; (**b**) GCD curves of the MHP nano-sheet arrays/ACF composite//AC hybrid device measured t within 0–1.4 V; (**a**,**c**) Ragone plot of the MHP nano-sheet arrays/ACF composite//AC hybrid devices and other phosphite materials reported in the literature.

## Data Availability

Data sharing not applicable.

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
