# Peer review of "Synthesis and Study on Ni-Co Phosphite/Activated Carbon Fabric Composited Materials with Controllable Nano-Structure for Hybrid Super-Capacitor Applications"

_nanomaterials, 2021, doi:10.3390/nano11071649_

Round 1
Reviewer 1 Report
Ni Co phosphate carbon composite electrode was synthesized via hydrothermal method and 3 electrode system was used to test the performance. However, similar structures with similar morphology was reported before. Some of them are
https://www.sciencedirect.com/science/article/pii/S1359836818342628
https://www.sciencedirect.com/science/article/pii/S1387700319312961
https://pubs.acs.org/doi/abs/10.1021/acsanm.0c02507?casa_token=V7t5kzCEFhsAAAAA:CNzYGNyZA6fQEnNakGN3IiPcobbS9uZ-GDnGGQg3iaIGtx2Q-I7wnDJStat6uqgSJbNJ8heWWSFRoQ
https://www.sciencedirect.com/science/article/pii/S0925838818304389?casa_token=2BQKkniQGVMAAAAA:KjSBdSBjPMrnRg4oUCs-N9MO4lFslTaBGShucPynCiNsOR6VHNE1xAxuONQajrVqYC6X-j8A
Some other comments are below
- There is no experimental results indicating flexibility. How do authors conclude the flexibility of the electrodes. Mechanical test should be reported.
- Details on asymmetrical cell was not provided.
- There is not enough explanation on electrochemical characterization
- How is the load density calculated?
- There is not enough discussion on characterization or electrochemical properties.
- For figure 5 a and b, why the order is different? the result does not make sense.
- The improvement with morphology was not explained.
- Novelty is not clear.
- The results were not explained .
- The results were not compared with previous reports.
- Two electrode results were not provided.
- Energy density and power density were not reported.
Author Response
To reviewer #1:
- Ni Co phosphate carbon composite electrode was synthesized via hydrothermal method and 3 electrode system was used to test the performance. However, similar structures with similar morphology was reported before. Some of them are
https://www.sciencedirect.com/science/article/pii/S1359836818342628
https://www.sciencedirect.com/science/article/pii/S1387700319312961
https://pubs.acs.org/doi/abs/10.1021/acsanm.0c02507?casa_token=V7t5kzCEFhsAAAAA:CNzYGNyZA6fQEnNakGN3IiPcobbS9uZ-GDnGGQg3iaIGtx2Q-I7wnDJStat6uqgSJbNJ8heWWSFRoQ
https://www.sciencedirect.com/science/article/pii/S0925838818304389?casa_token=2BQKkniQGVMAAAAA:KjSBdSBjPMrnRg4oUCs-N9MO4lFslTaBGShucPynCiNsOR6VHNE1xAxuONQajrVqYC6X-j8A
Thank you very much for providing the related papers to us. They are very helpful.
Reminded by another reviewer, the material obtained in our paper is called phosphite in English. Sorry for the mistake. We have corrected it in the revised manuscript.
Transition metal phosphites are a series of materials first reported in 1993, which is considered to have potential application values in many fields like ion-exchange and catalysis due to their open-framework structures [1]. Many groups reported the preparation of transition metal phosphites with certain morphologies, such as NiHPO3.H2O nano-needle bundles, Co11(HPO3)8(OH)6 with flower-like structures, Ni11(HPO3)8(OH)6 nano-rods [2-4]. However, a simple strategy for the synthesis of bi-metallic phosphites is still a challenge. What’s more, the preparation of phosphites arrays on substrate is a new challenge.Although, Ni/Co phosphites (MHP) nano-particles on conductive substrates such as carbon fabric, graphene, and carbon nanotube were reported to have superior electrochemical performance [5-7]. MHP arrays with controllable morphologies directly deposited on ACF by a simple hydrothermal method, which, to our knowledge, has not been reported by other groups. Interestingly, the MHP/ACF exhibits flexibility characteristic and shows attractive electrochemical performance as a binder free electrode for hybrid supercapacitor application.
- Marcos, M.; Amoros, P.; Porter, A.; Manez, R.; Attfield, J. Novel crystalline microporous transition-metal phosphites M11(HPO3)8(OH)6 (M = Zn, Co, Ni). X-ray powder diffraction structure determination of the cobalt and nickel derivatives. Chemistry of Materials 1993, 5, 121–128.
- Zhang, L.; Ni, Y.; Liao, K.; Wei, X. Large-scale synthesis of single crystalline NiHPO3· H2O nanoneedle bundles based on the dismutation of NaH2PO2. Crystal Growth and Design 2008, 8, 3636-3640.
- Ni, Y.; Liao, K.; Hong, J.; Wei, X. Ni2+ ions assisted hydrothermal synthesis of flowerlike Co11(HPO3)8(OH)6 superstructures and shape control. CrystEngComm 2009, 11, 570-575.
- Liao, K.; Ni, Y. Synthesis of hierarchical Ni11(HPO3)8(OH)6 superstructures based on nanorods through a soft hydrothermal route. Materials Research Bulletin 2010, 45, 205-209.
- Li, J.; Wang, Y.; Xu, W.; Wang, Y.; Zhang, B.; Luo, S.; Zhou, X.; Zhang, C.; Gu, X.; Hu, C. Porous Fe2O3 nanospheres anchored on activated carbon cloth for high-performance symmetric supercapacitors. Nano Energy 2019, 57, 379-387.
- Chen, J.; Xu, J.; Zhou, S.; Zhao, N.; Wong, C.-P. Template-grown graphene/porous Fe2O3 nanocomposite: a high-performance anode material for pseudocapacitors. Nano Energy 2015, 15, 719-728.
- Chen, L.-F.; Yu, Z.-Y.; Ma, X.; Li, Z.-Y.; Yu, S.-H. In situ hydrothermal growth of ferric oxides on carbon cloth for low-cost and scalable high-energy-density supercapacitors. Nano Energy 2014, 9, 345-354.
- Some other comments are below
- There is no experimental results indicating flexibility. How do authors conclude the flexibility of the electrodes? Mechanical test should be reported.
Figure Stress-strain curves of the MHP nano-sheet arrays/ACF composite;Tensile test was carried out to test the stress-strain deformation behavior of the composite. The results are presented in the revised manuscript (Figure 8a and SI-1).
- Details on asymmetrical cell was not provided.
In the revised paper, both three-electrode cell and two-electrode cell configurations were used. The electrochemical measurements of MHP/ACF composites electrodes were performed in a three-electrode cell (WE: MHP/ACF composites electrode; CE: a platinum foil; RE: Ag/AgCl/saturated KCl). 3 M KOH was used as electrolyte solution. For the hybrid device, the MHP/ACF composite was positive electrode, the ACF was negative electrodes, and electrolyte was1 M KOH.Corresponding details are presented in section: 2. Materials and Methods.
- There is not enough explanation on electrochemical characterization
A two-electrode electrochemical cell was fabricated to fully explore electrochemical performance of the MHP/ACF composite. Corresponding results and analysis are presented in the revised manuscript.
- How is the load density calculated?
Since the ACF substrate also contributes to electrochemical performance, it should not be ignored for the calculation of loading mass. Here in our manuscript, we use area density to consider all the active materials, A/cm2, referring to most reports using flexible substrates [1-3].
[1] Jien Li, et. al. Nano Energy 57 (2019) 379–387.
[2] Quan Zong, et. al.Chemical Engineering Journal 392 (2020) 123664.
[3] Amit Mishra, et. al. Chem. Electro. Chem. 6 (2019) 5771–5786.
- There is not enough discussion on characterization or electrochemical properties.
In this manuscript, we proposed and realized a facile one-step route to obtain Co/Ni phosphite arrays on ACF substrate. The composite materials form attractive binder free flexible electrodes for hybrid supercapacitor application. Electrochemical performance of materials can be affected by the morphology of MHP arrays, which can be easily tuned from bouquets to nano-sheets. A two-electrode electrochemical cell was fabricated to fully explore electrochemical performance of the MHP/ACF composite. Corresponding results and analysis are presented in the revised manuscript.
- For figure 5 a and b, why the order is different? The result does not make sense.
The mistakes in Figure 5 have been corrected.
- The improvement with morphology was not explained.
According to the experiment results, morphology of the MHP arrays highly depends on the concentration of NaH2PO2·H2O. We considered that the mixed solvent of water/DMAC is vital. It provides a basic environment to favor the dismutation of H2PO2− ions to HPO32−ions to finally form the phosphite [1]. NaH2PO2·H2O concentration controls the amount of nuclei and the growth rate of crystallite. The results in this manuscript are consistent with the usual crystallization rules. A large number of nuclei caused by high NaH2PO2·H2O concentration lead to the formation of nano-structure with sheet morphology. However, the explanation of the morphology improvement needs detailed research on crystal growth kinetics, crystal structure revolution analysis, and maybe some structural simulation calculation. Here in this manuscript, we want to emphasize the novelty of a facile one-step hydrothermal route to get the MHP/ACF composite with tunable morphology. Furthermore, the MHP/ACF composite materials exhibit excellent binder-free electrodes performance for flexible hybrid super-capacitor application. Based on the characterization and analysis, we explained the mechanism of the formation of characteristic morphologies and how solvent affected the morphology.
- Novelty is not clear.
To emphasize the novelty of this paper, introduction has been reconstructed. The novelty is briefly summarized as following:1. Providing a facile one-step route to obtain MHP arrays directly deposited on ACF substrate, where M=Co+Ni. It is a bi-metallic phosphite.2. Morphology of the MHP arrays can be easily tuned from bouquets to nano-sheets.3. The MHP/ACF composite materials form a binder free flexible electrode, which exhibit attractive electrochemical performance in hybrid supercapacitor.
- The results were not explained.
In conclusion section, we considered that: the unique nanostructure: nano-sheets directly grown on the activated carbon fabric (ACF), which has no additives and binders with higher specific surface areas and electrical conductivity, might be the major factor for the excellent electrochemical performance. The reasons of designing arrays on substrate for electrode application have also been added in the introduction section. Many reports considered that an ordered 1D nanostructure is particularly favored as an ideal structure for electrode materials because of its fast ion diffusion capability and large surface area.
- The results were not compared with previous reports.
A Ragone plot of the MHP/ACF composite//AC hybrid devices is given in Figure 9 (c), in which some reports are listed for comparison.
- Two electrode results were not provided.
A two-electrode results have been added in the revised manuscript. In the two-electrode device, the positive electrode is MHP nano-sheet arrays/ACF composite, negative electrode is ACF, and electrolyte is 1 M KOH.
- Energy density and power density were not reported.
A two-electrode electrochemical cell configuration was fabricated to fully explore the electrochemical performance of the MHP/ACF composite. The MHP/ACF composite@ACF hybrid supercapacitor can deliver an energy density of 34.1 Wh.kg-1 and power density of 722 W.kg-1 at 1 A.g-1. The results have been added in the revised manuscript.

Reviewer 2 Report
The authors report the synthesis, characterization, and performance of a nanostructured material M11(HPO3)8(OH)6 with the metal being Ni or Co. They claim that they could control the morphology between bouquet or sheet like nanostructures by controlling the amount of NaH2PO2.
They haven’t mentioned why they use specifically 2 or 4 mmol, what happens to the morphology at different concentrations?
The manuscript needs thorough English checking, especially grammar and formats.
Some examples: “binder-free electrodes candidates”, “ofMHP”, “peakscorrespond”, “flexiblehigh”
Line 65: They haven’t mentioned what is “DMA”.
Line 101: What kind of redox products are formed from dismutation of H2PO2- ?
For some reason, they haven’t used any names to describe many of the compounds in the manuscript. The synthesized materials are called transition-metal phosphites, which they haven’t mentioned anywhere.
Author Response
To reviewer #2:The authors report the synthesis, characterization, and performance of a nanostructured material M11(HPO3)8(OH)6 with the metal being Ni or Co. They claim that they could control the morphology between bouquet or sheet like nanostructures by controlling the amount of NaH2PO2.
- They haven’t mentioned why they use specifically 2 or 4 mmol, what happens to the morphology at different concentrations?
MHP/CNW composites with different morphologies were achieved by adding NaH2PO2·H2O with different concentration: the nano-bouquet arrays (2 mmol NaH2PO2·H2O) and the nano-sheet arrays (4 mmol NaH2PO2·H2O). The corresponding modification has been mentioned in section 2.2 in the revised manuscript.
- The manuscript needs thorough English checking, especially grammar and formats. Some examples: “binder-free electrodes candidates”, “ofMHP”, “peakscorrespond”, “flexiblehigh”
English checking has been thoroughly done in the revised manuscript.
- Line 65: They haven’t mentioned what is “DMA”.
We have added the full name of the solvent as Dimethylacetamide, and its abbreviation of DMAC has been corrected.
- Line 101: What kind of redox products are formed from dismutation of H2PO2-?
Usually, NaH2PO2 is used as a reductant, but it can also decompose into HPO32− and P3− ions due to the dismutation. According to Jin’s report Mn11(HPO3)8(OH)6 superstructures [1]. NiHPO3·H2O nano-needle bundles [2], Co11(HPO3)8(OH)6 superstructures [3] and Ni11(HPO3)8(OH)6 superstructures [4] were successfully prepared by employing NaH2PO2 as a phosphite source.
[1] L. Jin, J. Hong, Y. Ni, Mater. Chem. Phys. 123 (2010) 337.
[2] L. Zhang, Y.H. Ni, K.M. Liao, X.W.Wei, Cryst. Growth Des. 8 (2008) 3636.
[3] Y.H. Ni, K.M. Liao, J.M. Hong, X.W.Wei, Cryst. Eng. Comm. 4 (2009) 570.
[4] K.M. Liao, Y.H. Ni, Mater. Res. Bull. 45 (2010) 205.
- For some reason, they haven’t used any names to describe many of the compounds in the manuscript. The synthesized materials are called transition-metal phosphites, which they haven’t mentioned anywhere.
Thank you very much for this suggestion. We checked the whole manuscript and corrected the concepts of phosphate and phosphite. What’s more, we make it clear in the introduction part the synthesized Ni/Co phosphite on activated carbon fabrics.

Reviewer 3 Report
This paper describes the development and characterisation of binder-free electrodes for supercapacitor applications. It is a useful addition to the literature but a number of issues need to be addressed before it can be considered suitable for publication.
Firstly, this material is not strictly speaking a supercapacitor as it incorporates both capacitive and Faradaic charge storage. Hybrid supercapacitor would be a better term.
There are far too many significant figures (5) in the capacitance quoted in the Abstract, given the measurement uncertainty in this work. It should also be clarified in the Abstract that the results were obtained from three-electrode measurements.
A more complete description of the cotton fabric used in this work is required, in order to allow others to reproduce the results. Similarly, the conditions for ultrasonication of the precursor solution in the electrode synthesis step are missing.
The data in Figures 4-8 must be normalised to current density to facilitate comparison with other work.
Finally, the manuscript would benefit from thorough revision to correct a large number of typos and grammatical errors.
Author Response
To reviewer #3:This paper describes the development and characterisation of binder-free electrodes for supercapacitor applications. It is a useful addition to the literature but a number of issues need to be addressed before it can be considered suitable for publication.
- Firstly, this material is not strictly speaking a supercapacitor as it incorporates both capacitive and Faradaic charge storage. Hybrid supercapacitor would be a better term.
We very much agree with this comment. Corresponding revisions have been made in the title and the text.
- There are far too many significant figures (5) in the capacitance quoted in the Abstract, given the measurement uncertainty in this work. It should also be clarified in the Abstract that the results were obtained from three-electrode measurements.
We corrected the significant number to 4, referring to the test results of the similar electrochemical workstation [Jien Li, et. al. Nano Energy 57 (2019) 379–387.].The description of “three-electrode measurements” has been added in the abstract. We also applied the two electrode measurement results of a hybrid supercapacitor using MHP/ACF as positive electrode.
- A more complete description of the cotton fabric used in this work is required, in order to allow others to reproduce the results. Similarly, the conditions for ultrasonication of the precursor solution in the electrode synthesis step are missing.
The cotton fabric used was 100% cotton fabric purchased from the market (Hangzhou, Zhejiang, China). The ultrasonication of the precursor solution was carried out by a KH5200E ultrasonic cleaner (Hechuang Ultrasonic Instrument Co., Ltd, Kunshan, Jiangsu, China).The above description has been added in the revised manuscript.
- The data in Figures 4-8 must be normalised to current density to facilitate comparison with other work.
The data in Figure 4 and Figure 8 have been corrected to current density, A/cm2, referring to most reports using flexible substrates [1-3]. The mistakes of current in Figure 5 has been corrected.
[1] Jien Li, et. al. Nano Energy 57 (2019) 379–387.
[2] Quan Zong, et. al.Chemical Engineering Journal 392 (2020) 123664.
[3] Amit Mishra, et. al. Chem. Electro. Chem. 6 (2019) 5771–5786.
- Finally, the manuscript would benefit from thorough revision to correct a large number of typos and grammatical errors.
English checking has been thoroughly done in the revised manuscript.

Reviewer 4 Report
The synthesis of a new composite material of NiCi phosphate on a carbon support is presented. The claimed applicability of this material for super-capacitor applications is questionable because of limited capacity decrease. Nonetheless the reported results might be interesting from an academic point of view and the paper might be acceptable after the following points have been addressed:
Abstract:
14: of no high-resistivity -> of low resistivity
20: the abbreviation “CNW” is somewhat misleading as “CNX” is widely used for Carbon NanoX (X, e.g. tubes, fibers etc)
Introduction:
47: MHP has been hydrothermally synthesized by other groups, right (citation)? The novelty here is the synthesis together with the C fabric, I guess.
Materials:
57: Cotton fabric. Please specify.
Results:
90: Are these really nanowires? The nice SEM images reveal that some of the individual grains are compact while others are indeed bundles of small rods. A TEM investigation would be helpful for clarification.
112: The lattice parameters should be determined and compared with literature values. Does a solid solution series (Co,Ni911(HPO3)8(OH)6 exist?
124: The values given for the ratio Co+Ni/P are too accurate for EDS. Anyway, the ratio Ni/Co would be of more interest: does it vary or can it be varied by the synthesis?
169/222: The capacity loss of “nano-sheets” and “nano-bouquets” is de facto identical. Does it depend on the Ni:Co ratio? The manuscript fails to clearly work out the comparison of the good values reported in 222 to those in 169.
Author Response
To reviewer #4:The synthesis of a new composite material of NiCi phosphate on a carbon support is presented. The claimed applicability of this material for super-capacitor applications is questionable because of limited capacity decrease. Nonetheless the reported results might be interesting from an academic point of view and the paper might be acceptable after the following points have been addressed:
- Abstract: 14: of no high-resistivity -> of low resistivity
It has been revised as suggested.
- Abstract: 20: the abbreviation “CNW” is somewhat misleading as “CNX” is widely used for Carbon NanoX (X, e.g. tubes, fibers etc)
The abbreviation “CNW” has been changed to “ACF” (activated carbon fabrics).
- Introduction: 47: MHP has been hydrothermally synthesized by other groups, right (citation)? The novelty here is the synthesis together with the C fabric, I guess.
The novelty of this paper is the synthesis of MHP arrays directly on carbon fabric through a one-step route. The morphology of the MHP arrays can be tuned from bouquets to nano-sheets by an easy method. The binder free composited electrodes exhibit attractively excellent electrochemical performance.
- Materials: 57: Cotton fabric. Please specify.
The cotton fabric used was 100% cotton fabric purchased from the market (Hangzhou, Zhejiang, China). The description has been added in the revised manuscript.
- Results: 90: Are these really nanowires? The nice SEM images reveal that some of the individual grains are compact while others are indeed bundles of small rods. A TEM investigation would be helpful for clarification.
Here is the inset of Figure 1 (a). This is a hierarchical structure. Rods may be a more appropriate description for the primary structure, which assembles into bundles as the secondary structure. There are also a few rods existing individually. Since the SEM images give us enough information of the morphology, the TEM results have not been provided.
- Results: 112: The lattice parameters should be determined and compared with literature values. Does a solid solution series (Co,Ni911(HPO3)8(OH)6 exist?
XRD patterns show that both bouquet like and nano-sheet like arrays have similar diffraction patterns, which is in agreement with the monoclinic structure of Co11(HPO3)8(OH)6 (JCPDS NO. 81-1064) and Ni11(HPO3)8(OH)6 (JCPDS NO. 81-1065). Detailed analysis of the peak located at around 2θ=28.3° exhibits a slight shift of the diffraction peak. It is believed that Co and Ni ions co-exist in the lattice, where Co partially replaces Ni because of their very similar ionic radius.
- Results: 124: The values given for the ratio Co+Ni/P are too accurate for EDS. Anyway, the ratio Ni/Co would be of more interest: does it vary or can it be varied by the synthesis?
Significant figures of Co+Ni/P ratio have been corrected. As a result, the values are 1.5 and 1.4. According to the given experimental setup, Ni/Co ratio should be 1:1. EDS measurements reveal that the Ni/Co ratio is close to 1:1, which does not vary with the NaH2PO2·H2O concentration. As we believed, the ratio of Ni/Co can be controlled by the experimental setup. Researches on different ratio of Ni/Co should be further carried out in our future experiments.
- Results: 169/222: The capacity loss of “nano-sheets” and “nano-bouquets” is de facto identical. Does it depend on the Ni:Co ratio? The manuscript fails to clearly work out the comparison of the good values reported in 222 to those in 169.
Figure 5 c , Figure 7 b We don’t believe the capacity loss depends on the Ni:Co ratio in this work, since both the Ni:Co ratio in the samples are about 1:1.Results 169: The increased current density leads to a rapid decrease of specific capacity with capacity retention of 31.7% (nano-sheet arrays) and 28.5% (nano-bouquet arrays), respectively. The detailed results are listed in the following table, which was used to plot Figure 5 c. Here, both composited electrodes exhibit low rate capability.Results 222: Cyclic charge and discharge measurements were carried out at current density of 10 mA.cm-2. The specific capacity varied with the cycles was plotted in Figure 7 b. The specific capacity decreases from 1254 F·g-1 to 1057 F·g-1for the nano-sheet like arrays, 1155 F·g-1 to 952 F·g-1for the bouquet like arrays, after 1000 cycles. Both electrodes exhibit attractive cycling stability.

Round 2
Reviewer 3 Report
The authors have addressed most of my comments on the previous version of the manuscript. However, Figures 4-8 still show current (A) rather than current density (A/cm2) in the revised version. This must be corrected to facilitate comparison with other work.
Author Response
Corresponding revisions were made according to the reviewers suggestions.
